# Synergistic effect of probiotic, chicory root powder and coriander seed powder on growth performance, antioxidant activity and gut health of broiler chickens

Srinivas Gurram[1]*, V. Chinni Preetam[2], K. Vijaya Lakshmi[3], M. V. L. N. Raju[4], M. Venkateswarlu[5], Swathi Bora[6]

1 Poultry Research Station, PV Narsimha Rao Telangana Veterinary University, Rajendranagar, Hyderabad, 2 Department of Poultry Science, College of Veterinary Science, PV Narsimha Rao Telangana Veterinary University, Rajendranagar, Hyderabad, 3 Department of Livestock Farm Complex, PV Narsimha Rao Telangana Veterinary University, College of Veterinary Science, Mamnoor, Warangal, 4 Poultry Nutrition, ICAR, Directorate of Poultry Research, Rajendranagar, Hyderabad, India, 5 Department of Animal Nutrition, College of Veterinary Science, PV Narsimha Rao Telangana (_((((xxxVeterinary University, Rajendranagar, Hyderabad, 6 Department of Veterinary Pathology, College of Veterinary science, PV Narsimha Rao Telangana Veterinary University, Rajendranagar, Hyderabad

☯ These authors contributed equally to this work.
* gurramsrinivas4@gmail.com

**Data Availability Statement:** All data are fully available without restriction.

## Abstract

Gut health plays an important role on production and performance of broilers. This trial was undertaken with an aim to evaluate the synergistic effect of probiotic, chicory root powder and coriander seed powder on the performance and gut health of broiler chicken. For this purpose, a total of 240 day-old broiler chicks were randomly allotted to six dietary treatments with 8 replicates of 5 birds in each. Treatment groups included $T_1$ as control i.e., basal diet (BD) without any growth promoter and $T_2$—BD + antibiotic (BMD 0.05%). In the remaining experimental diets, $T_3$—probiotic (@ 0.01%) + chicory root powder (@ 1.0%), $T_4$—probiotic (@ 0.01%) + coriander seed powder (@ 1.5%), $T_5$—chicory root powder (@ 1.0%) + coriander seed powder (@ 1.5%) and $T_6$—probiotic (@ 0.01%) + chicory root powder (@ 1.0%) + coriander seed powder (@ 1.5%). The results indicated that supplementation of probiotic + chicory ($T_3$), probiotic + coriander ($T_4$), chicory + coriander ($T_5$) and probiotic + chicory + coriander ($T_6$) in combination resulted in significantly (P<0.05) higher weight gain and better FCR compared to control and antibiotic groups at 42 d of age. Supplementation of different dietary groups did not show any significant (P>0.05) effect on feed intake of broilers. Supplementation of all the test diets ($T_3$ to $T_6$) significantly (P<0.05) increased the glutathione peroxidase (GSHPx), glutathione reductase (GSHRx) and superoxide dismutase (SOD) enzyme activity when compared to control and antibiotic groups at 42 d of age. Supplementation of all the test diets ($T_3$ to $T_6$) significantly (P<0.05) lowered the pH in the gut, increased Lactobacillus counts, and reduced E. coli and Salmonella counts in the ileum compared to control and antibiotic groups. Supplementation of all the test diets ($T_3$ to $T_6$) significantly (P<0.05) increased the villus height (VH), crypt depth (CD), VH:CD ratio and villus width (VW) in the duodenum and only VH and CD in the ileum compared to control and antibiotic

**Funding:** The author(s) received no specific funding for this work.

**Competing interests:** The authors declare that they have no conflict of interest.

groups. Significantly (P<0.05) higher jejunal VH and VW and increased the goblet cell number in duodenum, jejunum and ileum was recorded in all test diets ($T_3$ to $T_6$) compared to control and antibiotic groups. Therefore, combinations of probiotic (0.01%), chicory root powder (1.0%) and coriander seed powder (1.5%) can be used as feed additive for improving performance and gut health of broiler chicken.

## Introduction

Antibiotics are being used as growth promoters in the poultry diets all over the world. However, in recent years, there has been rising demand to reduce or abolish the use of antibiotics as growth promoters due to the detrimental human health issue of antibiotic resistance [1]. Consumers' awareness of poultry products that do not contain antibiotic residues has increased, encouraging the use of suitable alternatives to antimicrobial compounds [2]. Among the feed additives, probiotics, prebiotics, organic acids, enzymes and medicinal plants have drawn more attention due to their prophylactic and growth promoting effects. Thus, the use of probiotics, medicinal plants, herbs and spices in poultry diets has become more popular worldwide as an alternative to antibiotics to minimize the disease incidence and achieving better performance in chicken [2].

Probiotics are single or mixed cultures of live microorganisms which beneficially affect the host by improving the balance of intestinal flora [3]. Probiotics maintain the beneficial intestinal microflora by competitive exclusion and antagonism, lowering the gut pH through acid fermentation, limiting the damage caused by pathogenic bacteria [4], improving epithelial cell integrity (villi height and width), producing bacteriocins, stimulating the gut associated immune system and increasing the production of short-chain fatty acids. Recently, herbal feed additive products like chicory root powder are gaining attention as they indirectly promote antimicrobial action by reducing the harmful bacteria in the gut. Dried chicory root powder is a good source of inulin type fructans and oligofructose chains known for having prebiotic action without any toxicity [5]. Inulin-type fructans are indigestible carbohydrates, recognized as dietary fibers that improve intestinal health and bird's performance through their prebiotic properties [6]. The fermentation activity of inulin inhibits the growth of harmful strains, selectively stimulates the growth of beneficial bacteria by decreasing the intestinal pH through increasing the absorption of short chain fatty acids and thus promotes the growth of broiler chickens [7, 8]. Similarly, probiotic + prebiotic supplementation decreased intestinal pH and viscosity in broilers [9]. Addition of probiotics (*Lactobacillus acidophilus and lactose fermenting enterobacteria)* and prebiotic combinations in broilers significantly (P<0.05) increased the villus height and crypt depth of the duodenum, ileum and jejunum [10].

Coriander (*Coriandrum sativum*) is regarded as both herb and spice, and has been used in medicine for thousands of years. Coriander seeds possess antioxidant, diuretic, anti-diabetic, hypocholesterolemic, antimicrobial, anthelmintic and anti-mutagenic qualities [11, 12]. Coriander seed powder contains 0.5–1.0% essential oil (carvone, geraniol, limonene, borneol, camphor, elemol and linalool) having antimicrobial properties against food borne pathogen such as *Salmonella* species [13]. In addition, it has appetizing and stimulatory effects in the digestion process by increasing production of digestive enzymes and juices, which stimulates digestion and peristaltic motion, thus improves feed efficiency [14, 15]. Coriander seed powder as an alternative to antibiotic growth promoter has been recommended for feeding in broilers by several authors [16–18]. In view of the above, this experiment was designed to evaluate the

synergistic effect of probiotic (0.01%), chicory root powder (1.0%) and coriander seed powder (1.5%) on the performance, antioxidant status and gut health of broiler chicken.

## Materials and methods

To conduct the study, two hundred and forty (240) day old commercial (Vencobb *400*) broiler chicks were procured, individually weighed and wing banded. The birds were distributed randomly into 6 dietary treatments, each with 8 replicates having 5 chicks in each replicate. The chicks were reared in battery brooders under standard managemental conditions. The experimental period was from day old to 42 d of age. The birds were fed with maize and soybean meal-based diets containing 2958, 3074 and 3163 kcal ME and 22.76, 21.58 and 19.68 percent crude protein, respectively during prestarter (0-14d), starter (15-28d) and finisher (28-42d) phases (Table 1). All the treatment concentrations in the feed were weight/weight basis. Treatment groups (Table 2) included $T_1$ as control i.e., basal diet (BD) without any growth promoter and $T_2$—BD + antibiotic (Bacitracin Methylene Disalicylate at 0.05%–manufacturer Zoetis). In the remaining experimental diets, $T_3$—probiotic (@ 0.01%) + chicory root powder (@ 1.0%), $T_4$—probiotic (@ 0.01%) + coriander seed powder (@ 1.5%), $T_5$—chicory root powder (@ 1.0%) + coriander seed powder (@ 1.5%) and $T_6$—probiotic (@ 0.01%) + chicory root powder (@ 1.0%) + coriander seed powder (@ 1.5%). The probiotic contains $10^9$ CFU/g of lyophilized and microencapsulated *Bacillus coagulans, saccharomyces boulardii, Lactobacillus acidophilus, Lactobacillus delbrueckii, Lactobacillus plantarum, Streptococcus thermophilus,*

**Table 1. Ingredient composition of basal diets (in kgs) fed to the commercial broilers from 0-42days.**

| Ingredient | Pre-starter (0-14d) | Starter (15-28d) | Finisher (29-42d) |
|---|---|---|---|
| Maize | 55.9 | 56.4 | 60.0 |
| Oil | 2.10 | 4.0 | 5.0 |
| Soyabean meal (CP 46%) | 37.1 | 34.8 | 30.1 |
| Stone grit | 1.58 | 1.83 | 1.88 |
| Dicalcium phosphate | 1.85 | 1.90 | 1.96 |
| Salt (NaCl) | 0.45 | 0.49 | 0.49 |
| DL-Methionine | 0.22 | 0.18 | 0.16 |
| L-Lysine HCl (99%) | 0.17 | 0.15 | 0.13 |
| Trace Mineral Mixture* | 0.10 | 0.10 | 0.10 |
| Vitamin AB2D3K** | 0.020 | 0.020 | 0.020 |
| Vitamin B-Complex*** | 0.025 | 0.025 | 0.025 |
| Choline chloride (50%) | 0.15 | 0.15 | 0.15 |
| Toxin binder | 0.10 | 0.10 | - |
| **Total** | **100** | **100** | **100** |
| Nutrient composition (calculated values) | | | |
| ME (kcal/kg) | 2964 | 3075 | 3167 |
| Crude protein (%) | 22.90 | 21.65 | 19.65 |
| Lysine (%) | 1.28 | 1.21 | 1.10 |
| Methionine (%) | 0.53 | 0.49 | 0.47 |
| Calcium (%) | 0.95 | 1.04 | 1.06 |
| Available phosphorous (%) | 0.45 | 0.45 | 0.45 |

*Trace mineral provided per kg diet (Avimin): Manganese 80mg, Zinc 70g, Iron 40g, Copper 8mg, Iodine 1gm and Selenium 0.25g.

**Vitamin AB2D3K each gram contains (Nicomix): Vitamin A 82500IU, Vitamin D3 12000IU, Vitamin B2 50mg, Vitamin K 10mg

*** Vitamin B-Complex each gram contains (Nicomix); Vitamin B1 4mg, Vitamin B6 8mg, Vitamin B12 40mcg, Niacin 60mg, Calcium Pantothenate 40mg, Vitamin E 40mg.

**Table 2. Experimental diets.**

| Trt | Experimental Diets |
|-----|---------------------|
| $T_1$ | Basal diet (BD) without additive |
| $T_2$ | BD + Antibiotic (BMD @ 0.05%) |
| $T_3$ | BD + Probiotic (@ 0.01%) + Chicory root powder (@ 1.0%) |
| $T_4$ | BD + Probiotic (@ 0.01%) + Coriander seed powder (@ 1.5%) |
| $T_5$ | BD + Chicory root powder (@ 1.0%) + Coriander seed powder (@ 1.5%) |
| $T_6$ | BD + Probiotic (@ 0.01%) + Chicory root powder (@ 1.0%) + Coriander seed powder (@ 1.5%) |

*Bacillus subtilis*, *Enterococcus faecium*, *Bifidobacterium bifidum*. The chemical composition of chicory root powder and coriander seed powder was given in Table 3. The dose levels (Probiotic at 0.01%, chicory at 1.0% and coriander at 1.5% levels) were selected based on previous trail results conducted.

Brooder temperature was maintained at 34 ± 1˚C up to 7 days of age and then gradually reduced to 26 ± 1˚C by 21 days of age after which chicks were maintained uniformly at room temperature. Feed and water were offered *ad libitum* throughout the experimental period. Weekly body weight, Feed intake and feed conversion ratios were calculated as feed intake per unit bodyweight gain at weekly intervals. The mortality rate was recorded throughout the experiment. The metabolic trial was conducted with one bird from each replicate to determine the retention efficiency of Dry Matter (DM), Crude Protein (CP) and energy as per the procedures described by AOAC (1997) [19]. The antioxidant enzymes such as glutathione Peroxidase (GSHPx), glutathione Reductase (GSHRx) and superoxide Dismutase (SOD) were estimated by following the methods of Paglia and Valentine (1967) [20], Carlberg and Mannervik (1985) [21] and Madesh and Balsubramanian (1998) [22] respectively.

Before slaughter, the birds were fasted overnight with free access to water and sacrificed by cervical dislocation and allowed for complete bleeding for 5 to 7 minutes. One bird from each replicate was sacrificed on 42nd day of age from each treatment group. Gut (proventriculus, gizzard, duodenum and ileum) pH was recorded immediately after collection of gastro-intestinal contents from respective part of gut. Approximately 1.0 g of sample content was suspended in 5ml distilled water, mixed vigorously with glass rod and pH was determined using digital pH meter. The electrode was rinsed with distilled water and recalibrated in between the readings [23].

## Gut ecology

Eight birds from each dietary treatment were slaughtered on 42nd day and intestines were dissected at Meckel's diverticulum. Approximately 5g of ileal digesta was collected aseptically into sterile sampling tubes and immediately transferred on ice to the laboratory for microbiological

**Table 3. Chemical composition of Chicory root powder.**

| Composition (%) | Chicory root powder | Coriander seed powder |
|-----------------|---------------------|------------------------|
| Moisture | 3.16 | 3.23 |
| Crude protein | 14.55 | 14.91 |
| Fat | 1.76 | 0.96 |
| Ash | 3.98 | 8.88 |
| Crude fiber | 30.01 | 34.21 |
| Total carbohydrates | 48.76 | 50.12 |
| Inulin | 46.89 | - |

examination for *E. coli*, *Salmonella* spp and *Lactobacilli* spp counts. Eosin methylene blue agar (EMB) was used for *E. coli* growth, Salmonella-Shigella agar (SS Agar) used for *Salmonella* spp. and MRS agar (De Man, Rogosa and Sharpe agar) used for *Lactobacilli* spp growth.

Then, 9 sterile test tubes with lids containing 9mL of phosphate buffer solution (PBS, pH-7.4) as diluent were prepared. Approximately 1g of the intestinal contents taken by sterile swab and homogenized for 3 min, aseptically mixed, added to the tubes, and diluted up to $10^9$. Later, 1ml of the contents of each test tube was transferred to one of three selective agar media on petri plates, respectively [24]. Aerobic bacterial plates (*E. coli*, *Salmonella* spp) were placed in an incubator at 37˚C for 24 hours. Anaerobic (*Lactobacilli* spp) medium plates were placed in an anaerobic jar with an anaerobic gas pack system at 37˚C for 24 hours. Finally, the intestinal bacterial colony populations formed in each plate was counted by colony counter and the number of colonies was expressed as log10 value.

## Histomorphometry

On $42^{nd}$ day during slaughter, 2 cm long segment of duodenum, jejunum and ileum of eight birds from each treatment were collected and then washed with physiological saline solution and fixed in 10% neutral buffered formalin solution. These samples were processed for histomorphological examination in terms of measurement of parameters like villous height (VH), cryptal depth (CD), villus width and villous height:crypt depth ratio. Histological technique involves processes like fixation of tissue, dehydration, clearing, embedding, cutting and staining. Fixation in 10% formalin with approximately 10–20 times the volume of the specimen was done. Tissues were dehydrated by using increasing strength of alcohol like 50%, 70%, 90% and 100%. Clearing was done by replacing alcohol by xylene for 0.5–1 hour. Impregnation of tissue with wax was done at melting point temperature of paraffin wax and the volume of wax was about 25–30 times the volume of tissues for a total duration of 4 hours. Impregnated tissues were placed in a mould with their labels and then fresh melted wax was poured in it and allowed to settle and solidify. These paraffin embedded tissues were sectioned at 5μm thickness and stained routinely with Hematoxylin-Eosin stain (H&E).

Histological sections were examined under 2X of light microscopy with micrometry and photographic attachment. The images were analyzed using image analyzing software (OLYMPUS cellSens Standard, version 1.13). A total of 20 intact well oriented crypt-villous units per bird were selected randomly, measured and the mean length was calculated for each sample. Villous height was measured from the tip of the villi to the base between individual villi, and crypt depth measurements were taken from the valley between individual villi to the basal membrane.

Record of temperature was maintained on daily basis where the highest daily average temperature recorded is 39.15˚C and the lowest temperature is 20.8˚C during the experimental period. The average relative humidity is 68.65 during the experimental period. The experiment was conducted during February and march- 2020.

Data analyzed for mean, standard errors and analysis of variance as per method of [25] and comparison of means were done [26] using software of Statistical Package for Social Sciences (SPSS) 20.0 version and significance was considered at P<0.05.

## Ethical approval

All authors hereby declare that all biological trials have been examined and approved by the ethics committee of PV Narsimha Rao Telangana Veterinary University, Rajendranagar, Hyderabad, India (Institutional Animal Ethics Committee number: IV/2019-02/IAEC/CVSC,

Hyderabad, India) and have therefore been performed in accordance with the ethical standards. No consent was raised by animal ethics committee while obtaining permission.

## Results and discussion

### Body weight gain

The results clearly indicated that supplementation of all test diets ($T_3$ to $T_6$) exhibited significantly ($P<0.05$) higher body weight gain compared to control ($T_1$) and antibiotic ($T_2$) groups at 42 days of age. The highest cumulative body weight gain (2185g) was recorded in probiotic + chicory ($T_3$) combination group followed by probiotic + coriander ($T_4$), chicory + coriander ($T_5$) and probiotic + chicory + coriander ($T_6$) groups. However, the lowest weight gain was recorded in control ($T_1$) and antibiotic ($T_2$) groups (Table 4). These results are in line with the findings of Taherpour *et al.* (2009) [27], who reported supplementation of probiotic and prebiotic combination improved the final body weight of broilers at 42 d of age. Similarly, supplementation of probiotic + prebiotic and probiotic + enzyme combination increased the body weight of broilers compared to control at 42 d of age [9]. Barad *et al.* (2017) [17] observed higher body weight gain in coriander seeds supplemented group when compared to control, turmeric powder and black pepper groups. Contrary to above results, Hofacre *et al.* (2003) [28] and Al-Khalaifa *et al.* (2019) [29] did not find positive effect on body weight in broilers fed with prebiotic + probiotic combination at 28 d of age.

The highest mean weight gain was recorded in probiotic + chicory ($T_3$) combination group which was significantly ($P<0.05$) higher among all the treatments. Similarly, Sanja *et al.* (2015) [30] reported addition of synbiotics (*Enterococcus faecim* + fructooligosaccharides) improved the body weight of broilers. Supplementation of probiotics and inulin combinations significantly ($P<0.05$) improved body weight gain in broilers [31]. The complimentary effect of probiotic and chicory powder on cumulative body weight gain as observed in the present study might be due to the suppression of undesirable microorganisms that lead to improved health status [32], increased nutrient digestibility, greater nutrient retention and improved gut health [33]. Similarly, increased body weight gains upon feeding diets containing probiotic + prebiotic combination [34–36] and probiotic + herb combination [37] in broiler chicken. Significant reduction in the counts of *E. coli* and *Salmonella* and reduction in gut pH by the supplementation of probiotic + chicory, probiotic + coriander, chicory + coriander and probiotic + chicory + coriander combinations in the present study is also in support with the authors. Contrary to above results, supplementation of probiotic + prebiotic combination did effect on body weight gain in broilers [10, 38, 39]

**Table 4. Synergistic effect of probiotic, chicory root powder and coriander powder on body weight gain (g), feed intake and feed conversion ratio of broiler chicken.**

| Trt | Diets | Body weight gain | Feed intake | Feed conversion ratio |
|---|---|---|---|---|
| $T_1$ | Control | 1975[d] | 3533 | 1.79[c] |
| $T_2$ | Antibiotic | 2016[c] | 3566 | 1.69[bc] |
| $T_3$ | Probiotic + Chicory | 2185[a] | 3553 | 1.63[a] |
| $T_4$ | Probiotic + Coriander | 2149[b] | 3564 | 1.66[ab] |
| $T_5$ | Chicory + Coriander | 2144[b] | 3561 | 1.66[ab] |
| $T_6$ | Probiotic + Chicory + Coriander | 2140[b] | 3545 | 1.65[ab] |
| | SEM | 10.627 | 15.869 | 0.0082 |
| | N | 8 | 8 | 8 |
| | *p*-value | 0.001 | 0.113 | 0.001 |

Value bearing different superscripts within a column are significantly ($P<0.05$) different.

## Feed Intake (FI)

The ANOVA revealed that there were no significant (P>0.05) differences in feed intake among different dietary treatments during overall experimental period (Table 4). The feed intake values at 42 d of age ranged between 3533 g to 3566 g. Similarly, supplementation of probiotic + prebiotic combination did not have significant (P>0.05) on FI of broilers [29]. In agreement with the results of this study, a series of earlier studies demonstrated that addition of probiotics + prebiotics [9, 36], chicory root powder [40, 41] and coriander seed powder [42] to the diet did not result in significant (P>0.05) effect on feed intake of broilers. On the contrary, probiotics + prebiotics [30], probiotic + herb combination [37], chicory root powder [30, 43] and coriander seed powder [17, 18] to the diets resulted in significant (P<0.05) effect on feed intake of broilers. These variations may be due to environmental factors and levels of the additives used in the experiment.

## Feed conversion ratio (feed intake/ body weight gain)

Supplementation of probiotic + chicory (T3), probiotic + coriander (T4), chicory + coriander (T5) and probiotic + chicory + coriander (T6) combination groups significantly (P<0.05) improved the efficiency of feed utilization compared to control and antibiotic. However, broilers fed with the probiotic + chicory (T3) combination group was more efficient at converting feed to body mass during entire experimental period (Table 4). To stimulate the growth of beneficial bacteria in the gut using a probiotic + chicory (T3) combination was more effective than the other combinations in this study. This might be due to symbiotic relation between chicory inulin and probiotic. Chicory root powder inulin serves as a source of nutrient for the probiotic bacterial cultures for early establishing in the gut. Similar results were reported by Szakacs *et al.* (2015) [44] and Sanja *et al.* (2015) [30] who stated that probiotic + prebiotic combination improved feed efficiency in broilers. Ashayerizadeh *et al.* (2009) [45] reported that addition of antibiotic, probiotic + prebiotic combination improved FCR compared to control. Improved feed efficiency with probiotic 0.4% + prebiotic 0.2% was also reported by Utami and Wahyono (2019) [36]. In agreement with the results of probiotic + coriander combination in this experiment, Hedayati and Manafi (2018) [37] reported that probiotic and herbal compound supplementation significantly (P<0.05) improved feed conversion ratio compared to control and antibiotic in broilers. However, in contrary to our findings, Kirkpinar *et al.* (2018) [9] and Al-Khalaifa *et al.* (2019) [29] did not find positive effect of probiotic and prebiotic combination on FCR of broilers.

Improvement in feed conversion efficiency in treatment groups might be attributed to enhanced digestive enzymes activity and an encouraged growth of the beneficial micro-flora in the GIT induced by dietary supplementation of probiotic, chicory root powder and coriander seed powder combination [46, 47]. Mode of action of above feed additives differs from one another, but in general they are all considered as antimicrobial agents. Improvement in feed efficiency might be obtained by several factors like alteration in intestinal pH, suppression of growth of intestinal pathogens, enhancement of growth of non-pathogenic bacteria and improvement of intestinal function (increased villi height, crypt depth and integrity) and nutrient digestibility.

## Nutrient utilization

Supplementation of all the test diets (T3 to T6) significantly (P<0.05) improved the energy retention, protein utilisation and dry matter digestibility compared to antibiotic, control groups (Table 5). The increased nutrient utilization in treatment groups might be due to probiotic bacteria, prebiotic properties of inulin and essential oils in coriander seed powder. Inulin-

**Table 5. Synergistic effect of probiotic, chicory root powder and coriander powder on nutrient utilization of broiler chicken.**

| Trt | Diets | Energy % | Protein % | Dry matter % |
|---|---|---|---|---|
| $T_1$ | Control | 70.52[c] | 80.11[c] | 72.65[c] |
| $T_2$ | Antibiotic | 72.25[b] | 82.06[b] | 74.18[b] |
| $T_3$ | Probiotic + Chicory | 75.77[a] | 84.94[a] | 76.89[a] |
| $T_4$ | Probiotic + Coriander | 75.52[a] | 84.16[a] | 76.80[a] |
| $T_5$ | Chicory + Coriander | 75.21[a] | 84.11[a] | 76.02[a] |
| $T_6$ | Probiotic + Chicory + Coriander | 75.68[a] | 84.01[a] | 76.92[a] |
| SEM | | 0.510 | 0.611 | 0.402 |
| N | | 8 | 8 | 8 |
| *p*-value | | **0.001** | **0.002** | **0.002** |

Value bearing different superscripts within a column are significantly (P<0.05) different

type fructan is a soluble fermentable fiber that is not digested by host digestive enzymes and serves as a substrate for beneficial like *bifidobacteriae* and *lactobacilli* in the lower part of the intestinal tract, the caeca and colon must be considered the sites of their effects on reabsorption of nutrients [48]. The mechanism by which inulin-type fructans may stimulate absorption is not well-known. The hypothesis more accepted is that the fermentation of inulin type fructans in the large intestine results in the production of short-chain fatty acids and lowers the gut pH. A lower intestinal pH facilitates absorption of nutrients [49, 50]. Similarly, Yang *et al.* (2008) [51] reported that supplementation of mannanoligosaccharides improved the energy and protein utilization in broilers. Justina *et al.* (2018) [52] indicated that supplementation of β-mannanase in broilers improved the dry matter digestibility and nutrient utilization. The enhanced dry matter digestibility and nutrient utilization may be attributed to the essential oils in coriander seed powder, which not only act as antibacterial and antioxidant, but also as stimulant of digestive enzymes in the intestinal mucosa, which might have improved the utilization of nutrients [15]. Similar results were also reported by Barad *et al.* (2017) [17] and Reddy *et al.* (2019) [53].

## Antioxidant enzyme activity

The glutathione peroxidase (Units/ml) enzyme activity was significantly (P<0.05) higher with all the test diets ($T_3$ to $T_6$) compared to control ($T_1$) and antibiotic ($T_2$), the highest enzyme activity being recorded in probiotic + chicory combination ($T_3$) and probiotic + coriander combination ($T_4$) groups. The other groups showed intermediate glutathione peroxidase enzyme activity. However, Supplementation of all test diet ($T_3$ to $T_6$) significantly (P<0.05) increased the glutathione reductase and superoxide dismutase enzyme activity compared to control and antibiotic (Table 6). Increased concentration of antioxidant enzymes in our study, is an indicator of better free radical scavenging of test diets. The steady state of antioxidant enzymes activity in all test groups may reflect a significant improvement in health and oxidative status of the birds. In agreement with the results, Tagang *et al.* (2013) [54] and Shen *et al.* (2014) [55] recorded increased (P<0.05) activity of serum catalase and glutathione peroxidase enzymes with probiotics in broilers. Similar results were also reported by Dong *et al.* (2019) [56] and Tengfei *et al.* (2019) [57] with probiotic supplementation in broilers.

In agreement with the positive results of chicory root powder on antioxidant activity, Sanja *et al.* (2015) [30] reported that addition of synbiotics (*Enterococcus faecim* + fructooligosaccharides) significantly (P<0.05) increased serum glutathione peroxidase, peroxidase, glutathione reductase and catalase enzyme activities compared to the control group. Similarly, Wang

**Table 6. Synergistic effect of probiotic, chicory root powder and coriander powder on antioxidant enzyme activity of broiler chicken at 42 d of age.**

| Trt | Diets | Glutathione peroxidase (Units/ml) | Glutathione reductase (Units/ml) | Superoxide dismutase (Units/mg protein) |
|---|---|---|---|---|
| $T_1$ | Control | 243[d] | 1636[b] | 6.62[b] |
| $T_2$ | Antibiotic | 314[c] | 1664[b] | 7.01[ab] |
| $T_3$ | Probiotic + Chicory | 397[a] | 1782[a] | 7.37[a] |
| $T_4$ | Probiotic + Coriander | 393[a] | 1792[a] | 7.25[a] |
| $T_5$ | Chicory + Coriander | 363[b] | 1814[a] | 7.24[a] |
| $T_6$ | Probiotic + Chicory + Coriander | 365[b] | 1835[a] | 7.39[a] |
| | SEM | 8.042 | 15.67 | 0.069 |
| | N | 8 | 8 | 8 |
| | *P*-value | 0.001 | 0.001 | 0.005 |

Value bearing different superscripts within a column are significantly (P<0.05) different

*et al.* (2018) [58] observed increased total antioxidant capacity in prebiotics than control and antibiotic (Aureomycin) in broilers. Decreased lipid peroxidation levels and increased activity of the superoxide dismutase and catalase enzymes in broilers fed with inulin was reported by Andreia *et al.* (2020) [59].

The increase in activity of these antioxidant enzymes with supplementation of probiotics and chicory root powder might be due to better control of intestinal pathogens in the gut. Aerobic bacteria (*Bacillus* spp) use oxygen in the intestine to provide an anaerobic environment for the colonization of anaerobic bacteria, such as *Lactobacilli* and *Bifidobacteria*. Therefore, these lactic acid-producing bacteria produce a more acidic environment, which impairs the growth of opportunistic pathogens [60]. *Lactobacillus acidophilus* increased the hydroxyl radical and hydrogen peroxide scavenging ability. Lactic acid bacteria could produce certain factors to capture reactive oxygen species (ROS) and prohibit the cytotoxic activity of ROS [61]. Significant reduction in the gut pH, *E. coli* and *Salmonella* count and increased in *Lactobacilli* count in all test diets also supported by the authors. Contrary to these findings, probiotics [54, 55], chicory powder [30] did not have any positive effect on antioxidant enzyme activity in broilers.

Chitra and Leelamma (1999) [11] demonstrated that coriander had a better antioxidative effect by increasing the activity of glutathione peroxidase, glutathione reductase and superoxide dismutase enzyme compared to control. Coriander is an egregious source of phyto-chemicals and functional compounds namely polyphenols, flavonoids and ascorbic acid which ultimately constitute for its high antioxidant activity. Darughe *et al.* (2012) [62] demonstrated that essential oil of coriander contains camphor, cyclohexanol acetate, limonene, α-pinene and inhibited the rate of primary and secondary oxidation products formation and their effects were almost equal to BHA. The improvement in the antioxidant enzyme activity observed with the addition of coriander seed powder could be attributed to the presence of essential oils and their main components, linalool, trepene and terpenoid [63].

## Gut pH

Supplementation of all test diets ($T_3$ to $T_6$) significantly (P<0.05) lowered the pH in duodenum, jejunum, ileum and caecum (except proventriculus) compared to control and antibiotic groups (Table 7). Probiotic bacteria produce short chain acids like lactic, acetic and other organic acids, which are responsible for reduction in the intestinal pH [64]. Aerobic bacteria (*Bacillus* spp) use oxygen in the intestine to provide an anaerobic environment for the colonization of anaerobic bacteria, such as *Lactobacilli* and *Bifidobacteria*. Therefore, these lactic

**Table 7. Synergistic effect of probiotic, chicory root powder and coriander powder on gut pH of broiler chicken at 42 d of age.**

| Trt | Diets | Proventriculus | Duodenum | Jejunum | Ileum | Caecum |
|---|---|---|---|---|---|---|
| T1 | Control | 3.54 | 6.11$^c$ | 6.43$^d$ | 6.66$^c$ | 7.39$^c$ |
| T2 | Antibiotic | 3.56 | 6.01$^b$ | 6.30$^c$ | 6.45$^b$ | 7.18$^b$ |
| T3 | Probiotic + Chicory | 3.51 | 5.89$^a$ | 6.06$^a$ | 6.23$^a$ | 6.94$^a$ |
| T4 | Probiotic + Coriander | 3.48 | 5.88$^a$ | 6.15$^{ab}$ | 6.30$^a$ | 6.95$^a$ |
| T5 | Chicory + Coriander | 3.54 | 5.94$^{ab}$ | 6.20$^b$ | 6.31$^a$ | 6.99$^a$ |
| T6 | Probiotic + Chicory + Coriander | 3.54 | 5.94$^{ab}$ | 6.18$^b$ | 6.25$^a$ | 7.03$^a$ |
| | SEM | 0.0098 | 0.0155 | 0.0208 | 0.0270 | 0.0301 |
| | N | 8 | 8 | 8 | 8 | 8 |
| | *p*-value | **0.157** | **0.001** | **0.001** | **0.001** | **0.001** |

Value bearing different superscripts within a column are significantly (P<0.05) different

acid-producing bacteria produce a more acidic environment, which impairs the growth of opportunistic pathogens [60]. Significant increase in the counts of *Lactobacilli* in test diets also support the above results. Al-Khalaifa *et al.* (2019) [29] reported that supplementation of probiotics and prebiotics driven the gut pH value towards acidity, but it failed to reach significance, as it was only a numerical difference. Whereas, Denli *et al.* (2003) [65] reported that inclusion of probiotic at 0.1% and antibiotic at 0.15% in broiler diets did not have any effect on the intestinal pH.

## Gut ecology

Supplementation of all the test diets (T$_3$ to T$_6$) including antibiotic group significantly (P<0.05) decreased the *E. coli* counts compared to control (Table 8). The lowest *E. coli* counts were recorded in antibiotic group (T$_2$), probiotic + chicory (T$_3$) and probiotic + chicory + coriander (T$_6$) groups followed by probiotic + coriander (T$_4$) and chicory + coriander (T$_5$) groups. Supplementation of antibiotic (T$_2$) significantly (P<0.05) decreased the ileal *Salmonella* counts compared to control and other test diets. The *Salmonella* count in probiotic + chicory + coriander (T$_6$), probiotic + coriander (T$_4$) groups and probiotic + chicory (T$_3$) showed intermediate values, but they had significantly (P<0.05) lower *Salmonella* counts than the control (T$_1$) and chicory + coriander (T$_5$) groups. In agreement with the above results, Karwan *et al.* (2016)

**Table 8. Synergistic effect of probiotic, chicory root powder and coriander powder on gut microbiota (log$_{10}$ of cfu/g count) in ileum sample of broiler chicken.**

| Trt | Diets | *Escherichia coli* (log$_{10}$ cfu/g) * | *Salmonellea* spp. (log$_{10}$ cfu/g) ** | *Lactobacillus* spp. (log$_{10}$ cfu/g) * |
|---|---|---|---|---|
| **T$_1$** | **Control** | 8.29$^d$ | 4.23$^d$ | 7.89$^c$ |
| **T$_2$** | **Antibiotic** | 7.15$^a$ | 3.23$^a$ | 6.93$^d$ |
| **T$_3$** | **Probiotic + Chicory** | 7.18$^a$ | 3.92$^c$ | 8.08$^a$ |
| **T$_4$** | **Probiotic + Coriander** | 7.33$^b$ | 3.90$^c$ | 7.86$^c$ |
| **T$_5$** | **Chicory + Coriander** | 7.42$^c$ | 4.16$^d$ | 7.88$^c$ |
| **T$_6$** | **Probiotic + Chicory + Coriander** | 7.19$^a$ | 3.81$^b$ | 7.94$^b$ |
| | SEM | 0.059 | 0.049 | 0.055 |
| | N | 8 | 8 | 8 |
| | *P*-value | **0.001** | **0.001** | **0.001** |

Values bearing different superscripts within a column are significantly (P<0.05) different

* Calculated as per log$_{10}$ colony forming units/gram of sample (10$^6$).

** Calculated as per log$_{10}$ colony forming units/gram of sample (10$^3$).

**Table 9. Synergistic effect of probiotic, chicory root powder and coriander powder on histomorphometry of duodenum of broiler chicken.**

| Trt | Diets | Villus height (µm) | Crypt depth (µm) | Villus height: Crypt depth Ratio | Villus width (µm) |
|---|---|---|---|---|---|
| $T_1$ | Control | 1025.84[c] | 240.58[c] | 4.29[c] | 134.11[c] |
| $T_2$ | Antibiotic | 1174.89[b] | 260.43[c] | 4.55[b] | 156.05[bc] |
| $T_3$ | Probiotic + Chicory | 1598.52[a] | 315.58[b] | 5.15[a] | 213.64[a] |
| $T_4$ | Probiotic + Coriander | 1261.81[b] | 308.45[b] | 4.10[d] | 171.78[b] |
| $T_5$ | Chicory + Coriander | 1623.28[a] | 319.91[ab] | 5.07[a] | 168.11[b] |
| $T_6$ | Probiotic + Chicory + Coriander | 1651.62[a] | 343.38[a] | 4.87[ab] | 199.69[a] |
| | SEM | 43.420 | 6.825 | 0.087 | 5.764 |
| | N | 8 | 8 | 8 | 8 |
| | *P*-value | **0.001** | **0.001** | **0.001** | **0.001** |

Values bearing different superscripts within a column are significantly (P<0.05) different

[31] observed that addition of postbiotics and inulin combinations significantly (P<0.05) reduced the *Enterobacteriaceae* count compared to control. Similarly, Biswas *et al.* (2018) [8] reported that supplementation of antibiotics (BMD) and prebiotics (MOS and FOS) reduced the total anaerobes and *coliforms* counts in ileum of broilers. Supplementation of antibiotic, probiotic and herbal compound significantly (P<0.05) reduced the *E. coli*, *Salmonella* and *coliforms* counts compared to control in broilers was reported by Hedayati and Manafi (2018) [37]. Probiotic bacteria produce short chain acids which decreases the intestinal pH and encourages the growth of Lactobacilli and *Bifidobacteria* [64]. Therefore, these lactic acid-producing bacteria produce a more acidic environment, which impairs the growth of opportunistic pathogens [60]. Significant increase in *Lactobacilli* count in all test diets was also support the above results. On contrary, Wang *et al.* (2018) [58] did not find any significant (P>0.05) difference in total anaerobic bacterial count in broilers with probiotic and antibiotic supplementation.

In agreement with the lowered *E. coli* and *Salmonella* counts in probiotic + coriander ($T_4$), chicory + coriander ($T_5$) groups, Ghazanfari *et al.* (2015) [66] reported that supplementation of antibiotic and coriander oil lowered the caecal *E. coli* counts than control in broilers. Similarly, Taha *et al.* (2019) [18] observed that coriander seed powder decreased the total bacterial, *E. coli* and *C. perfringens* counts in the ileum of broilers. The decreased pathogenic bacterial load in the ileum might be due to essential oils in coriander have which hydrophobic

**Table 10. Synergistic effect of probiotic, chicory root powder and corianderpowder on histomorphometry of jejunum of broiler chicken.**

| Trt | Diets | Villus height (µm) | Crypt depth (µm) | Villus height: Crypt depth Ratio | Villus width (µm) |
|---|---|---|---|---|---|
| $T_1$ | Control | 949.13[c] | 171.25 | 5.68 | 153.23[b] |
| $T_2$ | Antibiotic | 1010.44[b] | 168.96 | 5.99 | 164.00[b] |
| $T_3$ | Probiotic + Chicory | 1082.86[a] | 165.90 | 6.69 | 210.44[a] |
| $T_4$ | Probiotic + Coriander | 1013.05[b] | 171.70 | 5.93 | 197.99[a] |
| $T_5$ | Chicory + Coriander | 1044.34[ab] | 167.88 | 6.26 | 204.35[a] |
| $T_6$ | Probiotic + Chicory + Coriander | 1061.79[ab] | 175.81 | 6.09 | 207.68[a] |
| | SEM | 10.135 | 1.672 | 0.106 | 4.257 |
| | N | 8 | 8 | 8 | 8 |
| | *P*-value | **0.001** | **0.638** | **0.107** | **0.001** |

Values bearing different superscripts within a column are significantly (P<0.05) different

**Table 11. Synergistic effect of probiotic, chicory root powder and coriander powder on histomorphometry of ileum of broiler chicken.**

| Trt | Diets | Villus height (µm) | Crypt depth (µm) | Villus height: Crypt depth Ratio | Villus width (µm) |
|---|---|---|---|---|---|
| $T_1$ | Control | 562.15[d] | 172.87[b] | 3.28[c] | 131.15 |
| $T_2$ | Antibiotic | 668.51[c] | 158.67[b] | 4.17[b] | 129.05 |
| $T_3$ | Probiotic + Chicory | 758.38[b] | 160.25[b] | 4.63[b] | 136.76 |
| $T_4$ | Probiotic + Coriander | 1019.99[a] | 199.06[a] | 5.22[a] | 145.61 |
| $T_5$ | Chicory + Coriander | 752.38[b] | 166.24[b] | 4.47[b] | 131.87 |
| $T_6$ | Probiotic + Chicory + Coriander | 742.35[b] | 193.22[a] | 3.83[c] | 134.18 |
| | SEM | 25.265 | 4.294 | 0.116 | 2.246 |
| | N | 8 | 8 | 8 | 8 |
| | *P*-value | **0.001** | **0.001** | **0.001** | **0.336** |

Values bearing different superscripts within a column are significantly (P<0.05) different

properties [67] that affect cell wall lipids of the bacteria by disturbing bacterial structures and rendering them more permeable, thus results in lower number of harmful bacteria.

The *Lactobacilli* counts were significantly (P<0.05) increased with the supplementation of probiotic + chicory ($T_3$) followed by probiotic + chicory + coriander ($T_6$) when compared to other treatment groups. Birds supplemented with antibiotic in the diet showed significantly (P<0.05) lower *Lactobacilli* counts. However, no significant (P>0.05) difference was recorded among control ($T_1$), probiotic + coriander ($T_4$) and chicory + coriander ($T_5$) groups. Similarly, Dong *et al.* (2019) [56] reported that supplementation of microencapsulated probiotics significantly (P<0.05) increased the *Lactobacilli* counts in caecum of broilers. Similarly, Biswas *et al.* (2018) reported that supplementation of antibiotics (BMD) and prebiotics (MOS and FOS) increased the *Lactobacilli* counts in ileum of broilers. Increased faecal lactic acid bacteria in

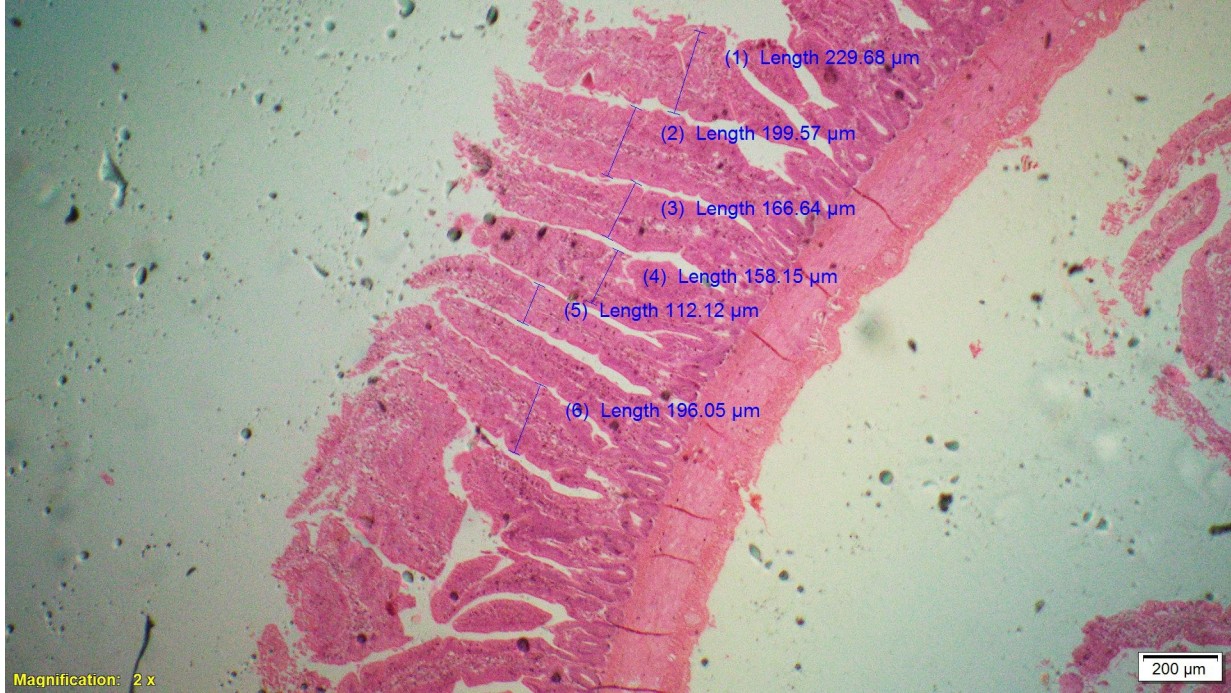

**Fig 1. Photomicrograph of the cross section of Jejunum from control group (T1).** H&E, 2x.

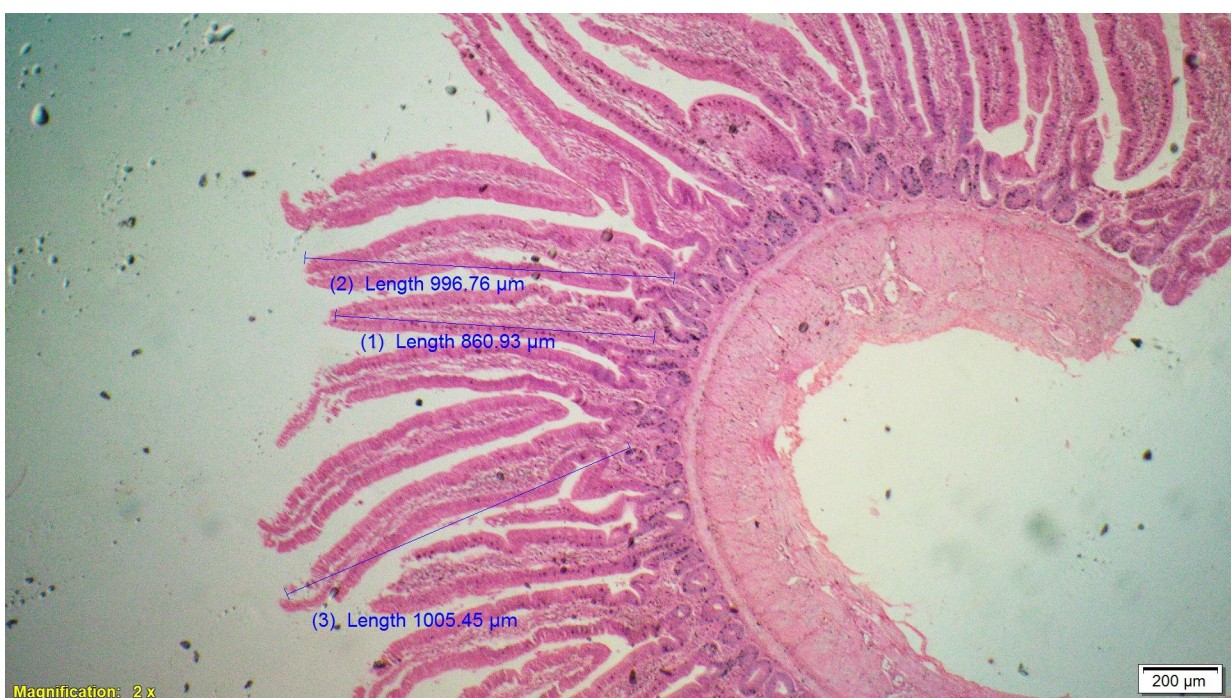

**Fig 2. Photomicrograph of the cross section of Jejunum from antibiotic group (T2).** H&E, 2x.

postbiotics and inulin combination groups was also reported by Karwan *et al*. (2016) [31] in broilers. Wang *et al*. (2018) [58] observed that microencapsulated probiotics and prebiotics

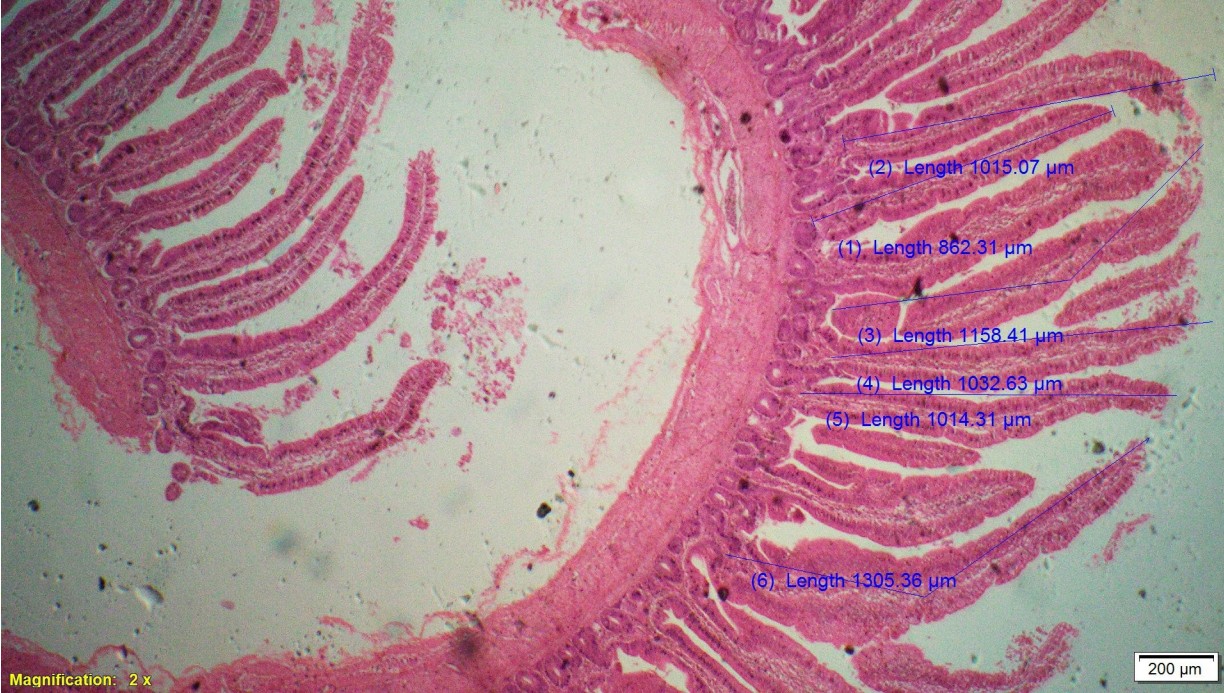

**Fig 3. Photomicrograph of the cross section of Jejunum from probiotic + chicory group (T3), H&E, 2x.**

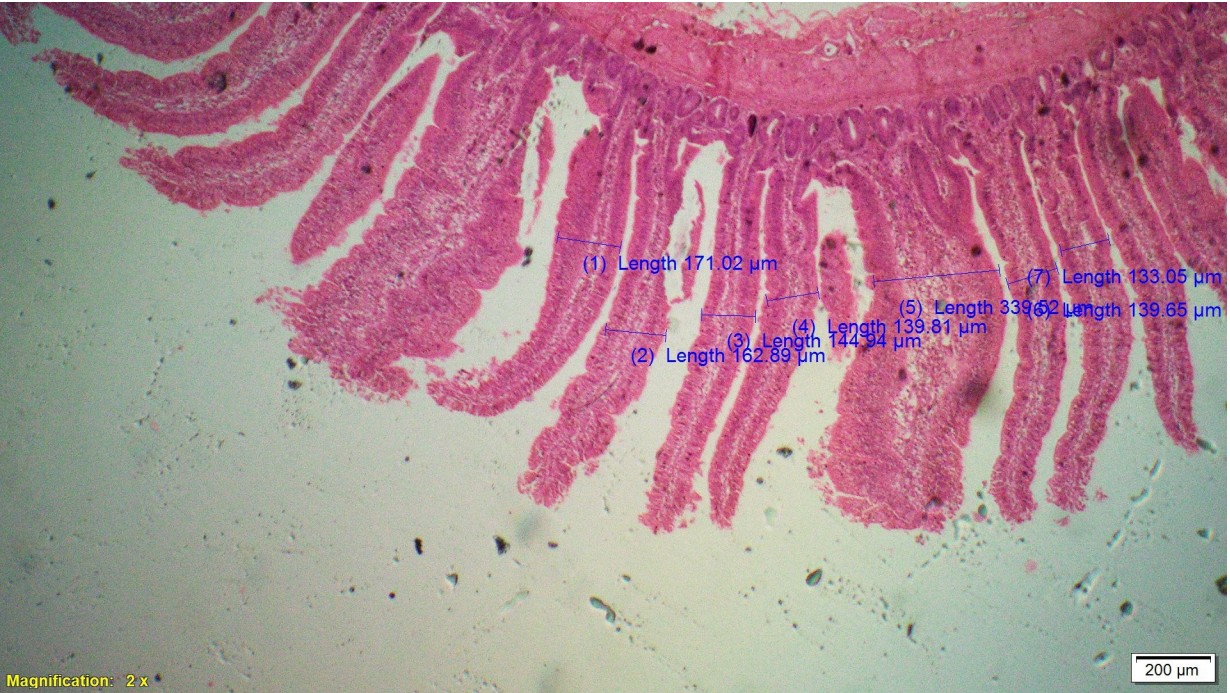

**Fig 4. Photomicrograph of the cross section of Jejunum from probiotic + coriander group (T4), H&E, 2x.**

(MEP) significantly (P<0.05) increased the *Lactobacilli* counts than control and antibiotic in broilers. In contrary, supplementation of probiotic did not have significant (P<0.05) effect in cecal *Lactobacilli* counts of broilers [29].

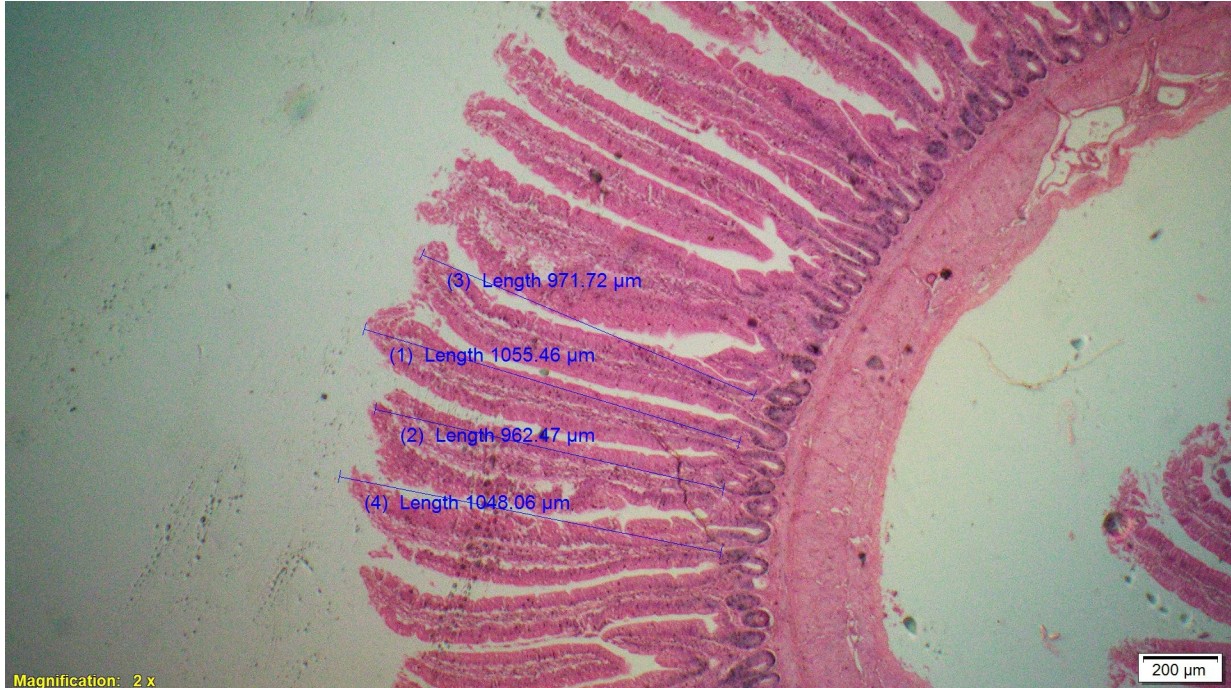

**Fig 5. Photomicrograph of the cross section of Jejunum from chicory + coriander group (T5), H&E, 2x.**

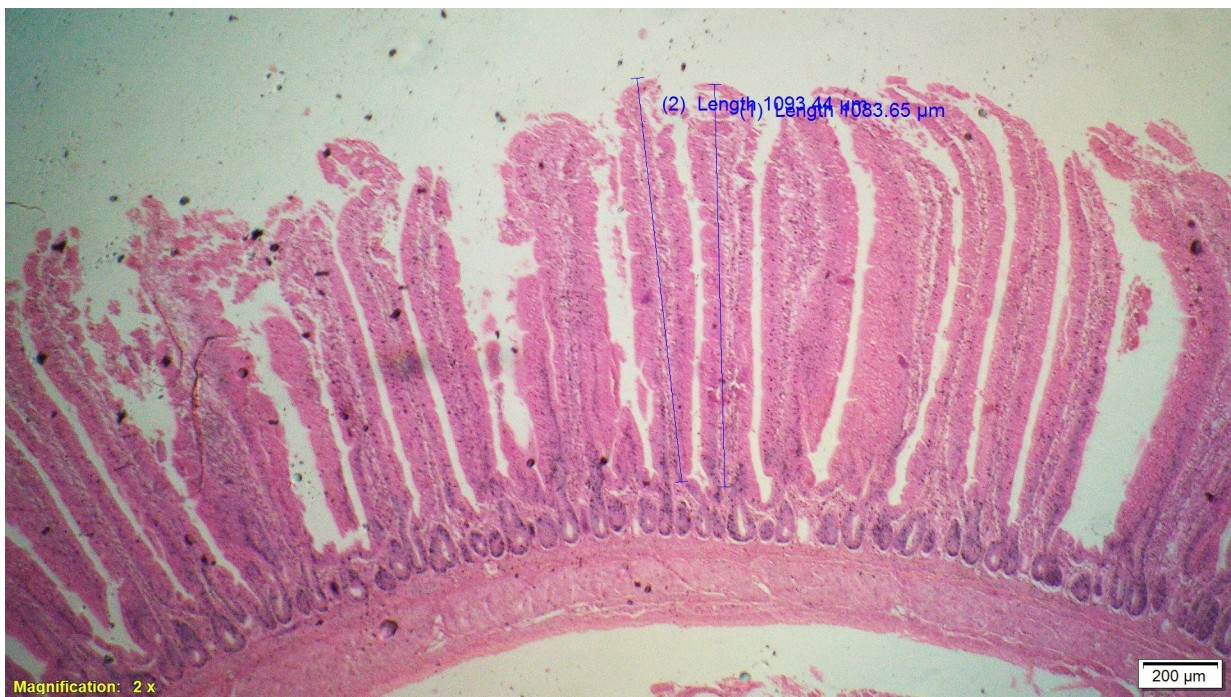

**Fig 6. Photomicrograph of the cross section of Jejunum from probiotic + chicory + coriander group (T6), H&E, 2x.**

The increased *Lactobacilli* count in probiotic + chicory ($T_3$) combination group might be due to probiotic bacteria such as *Lactobacillus* spp. or *Bifidobacterium* spp. use inulin for fermentation more efficiently than other groups of bacteria and produces short chain fatty acids on inulin to create an acidic environment which suppresses the growth of acid intolerant bacteria like *Salmonellae* and *E. coli* and enhanced the growth of acid tolerant bacteria like *Lactobacilli* and *Bifidobacterium* [68].

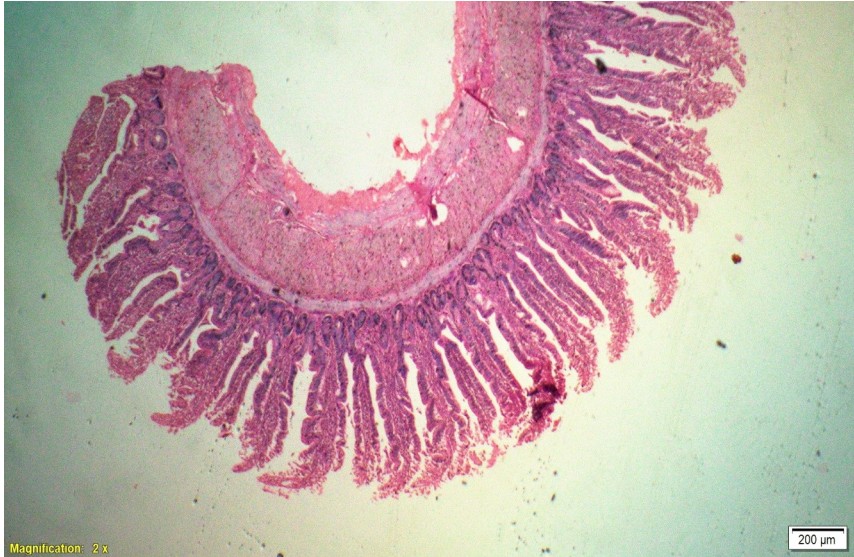

**Fig 7. Photomicrograph of the cross section of ileum from control group (T1).** H&E, 2x.

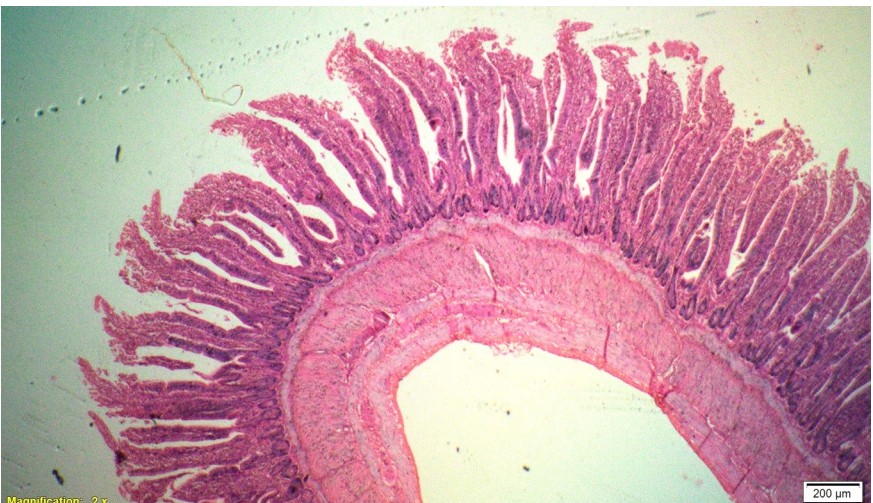

**Fig 8. Photomicrograph of the cross section of ileum from antibiotic group (T2).** H&E, 2x.

## Gut histomorphometry

Supplementation of all the test diets ($T_3$ to $T_6$) significantly (P<0.05) increased the villus height (VH), crypt depth (CD), VH:CD ratio and villus width (VW) in the duodenum and only VH and CD in the ileum compared to control and antibiotic groups. Significantly (P<0.05) higher jejunal VH and VW was recorded in all test diets compared to control and antibiotic groups. Increased villus height and villus width enhances the absorptive capacity of the small intestine by reducing the digesta passage rate and therefore, optimize broiler performance. However, supplementation of different dietary groups did not show any significant (P>0.05) effect on jejunal CD and VH:CD ratio and ileal villus width at 42 d of age (Tables 9– 11 and Figs 1–12).

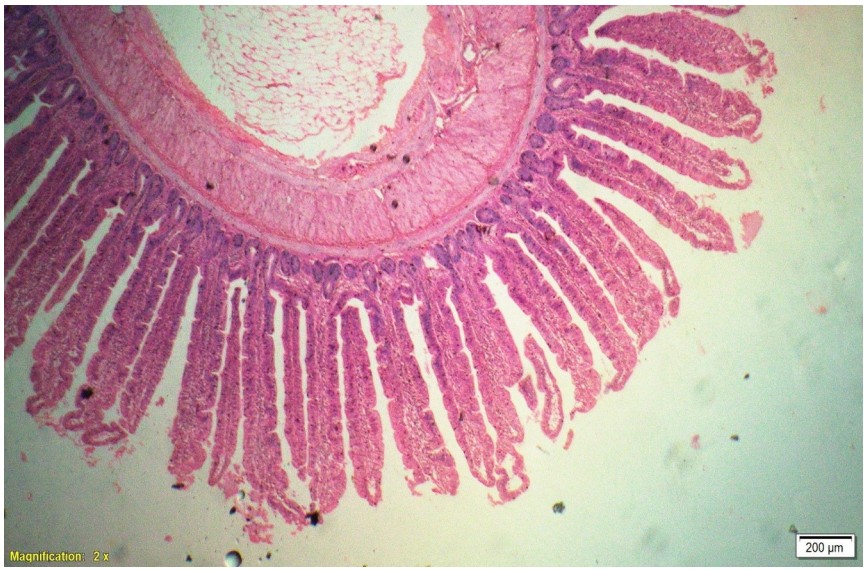

**Fig 9. Photomicrograph of the cross section of ileum from probiotic + chicory group (T3), H&E, 2x.**

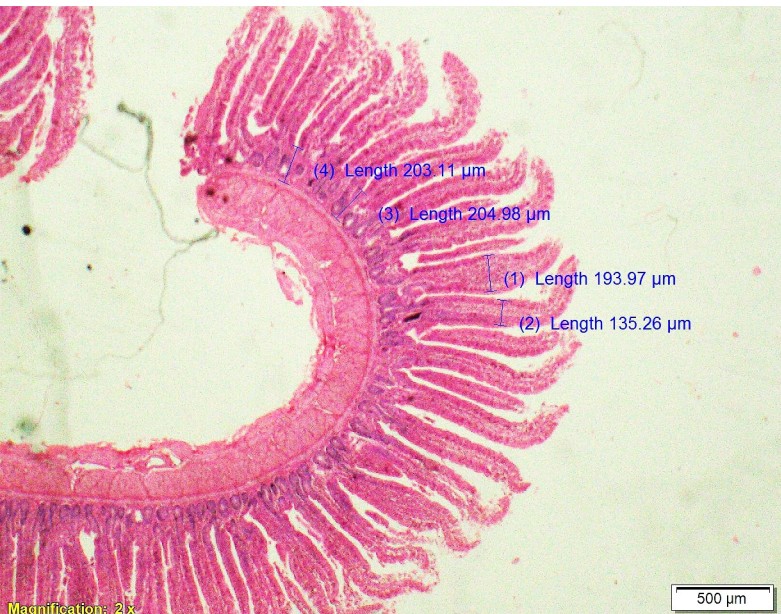

**Fig 10. Photomicrograph of the cross section of ileum from probiotic + coriander group (T4), H&E, 2x.**

In agreement with the above results, Karwan *et al.* (2016) [31, 69] reported that addition of probiotics and inulin combinations increased the villus height and crypt depth of the duodenum, ileum and jejunum compared to control. Similarly, supplementation of probiotic + organic acid combination significantly (P<0.05) increased the duodenal VH, CD and VH: CD in broilers [70]. The positive effect of probiotics and chicory root powder on the intestinal morphology mainly arose from its ability to create a favourable intestinal environment which had a better effect on intestinal morphology [71]. Probiotics and chicory root powder increase to production of short chain fatty acids and reduce intestinal pH. Hence, beneficial effects on intestinal tissue health and morphology are achieved. In contrary, Fernandes *et al.* (2014) [10]

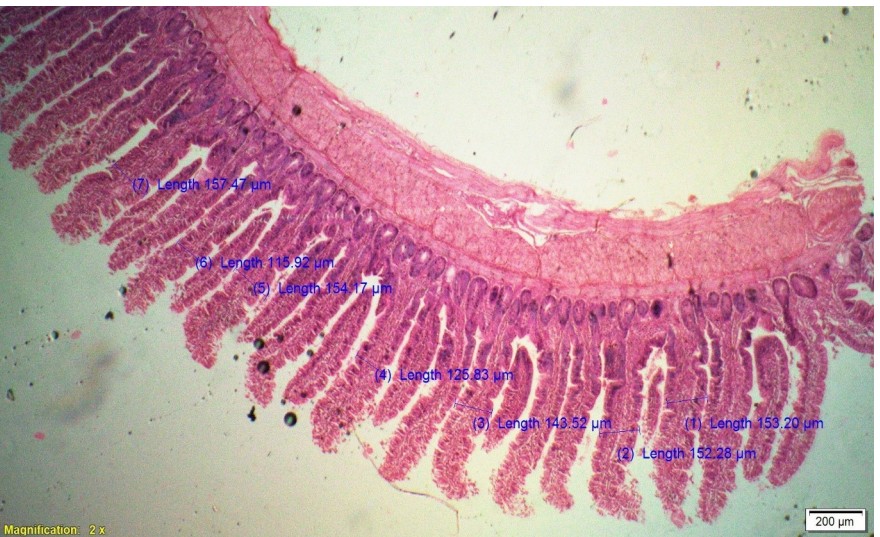

**Fig 11. Photomicrograph of the cross section of ileum from chicory + coriander group (T5), H&E, 2x.**

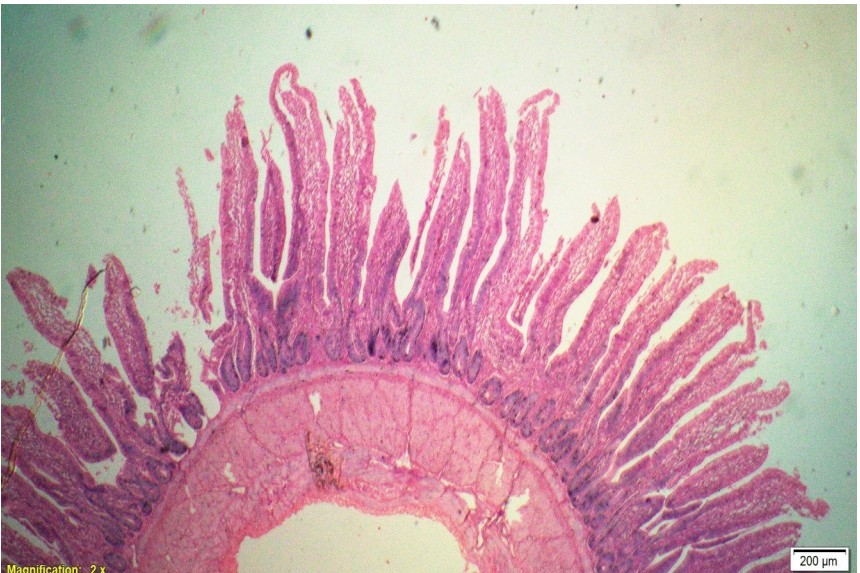

**Fig 12. Photomicrograph of the cross section of ileum from probiotic + chicory + coriander group (T6), H&E, 2x.**

reported that supplementation of probiotic + prebiotic combination did not have any significant (P>0.05) effect on intestinal integrity of broiler.

Probiotic, chicory root powder and coriander seed powder may reduce the growth of many pathogenic and non-pathogenic intestinal bacteria thereby resulting in reduction in intestinal colonization and infectious process which ultimately decrease the inflammatory process of intestinal mucosa resulting in improved villus height and villus width which in turn increases secretory function, digestion, and absorption of nutrients [8]. It is hypothesised that the increase in beneficial microbial activity resulting from dietary probiotic, chicory root powder and coriander seed powder supplementation may influence gut morphology and consequently affect gut maturation.

## Conclusion

Supplementation of probiotic + chicory, probiotic + coriander, chicory + coriander and probiotic + chicory + coriander combinations produced greater weight gain, improved FCR, and higher antioxidant activity compared to control and antibiotic at 42 d of age. The combination of probiotic (0.01%) with chicory root powder (1.0%) was more effective than combinations of other additives in terms of body weight gain and FCR. Supplementation of different combinations of probiotic, chicory root powder and coriander seed powder significantly lowered the gut pH, E. coli and Salmonella counts and increased the Lactobacilli counts. In addition, this treatment improved the gut morphometry parameters such as VH, CD and VW in the in the intestines. Thus, supplementation of probiotic at 0.01%, chicory root powder at 1.0%, and coriander seed powder at 1.5% combinations could be used in the diet as a potential growth promoter in broiler chickens. However, follow up large-scale studies under field conditions are necessary before recommending the compounds in the broiler diet.

## Supporting information

**S1 Data.**
(XLSX)

## Acknowledgments

The presented manuscript is a part of the first Author's PhD dissertation. The authors are thankful to Department of Poultry Science, College of Veterinary science, PV Narsimha Rao Telangana Veterinary University, R'nagar, Hyderabad, India.

## Author Contributions

**Conceptualization:** Srinivas Gurram, K. Vijaya Lakshmi, M. V. L. N. Raju, M. Venkateswarlu.

**Data curation:** Srinivas Gurram.

**Formal analysis:** Srinivas Gurram.

**Funding acquisition:** Srinivas Gurram.

**Investigation:** Srinivas Gurram.

**Methodology:** Srinivas Gurram, M. V. L. N. Raju, Swathi Bora.

**Project administration:** Srinivas Gurram.

**Resources:** Srinivas Gurram.

**Software:** Srinivas Gurram.

**Supervision:** Srinivas Gurram, V. Chinni Preetam, K. Vijaya Lakshmi, M. V. L. N. Raju, M. Venkateswarlu.

**Validation:** Srinivas Gurram.

**Visualization:** Srinivas Gurram.

**Writing – original draft:** Srinivas Gurram.

**Writing – review & editing:** Srinivas Gurram, M. V. L. N. Raju, Swathi Bora.

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
