## [Decision Letter · Decision Letter 0]

7 Mar 2022

PONE-D-21-40040Synergistic effect of probiotic, chicory root powder and coriander seed powder on growth performance, antioxidant activity and gut health of broiler chickenPLOS ONE

Dear Dr. Gurram,

Thank you for submitting your manuscript to PLOS ONE. After careful consideration, we feel that it has merit but does not fully meet PLOS ONE’s publication criteria as it currently stands. Therefore, we invite you to submit a revised version of the manuscript that addresses the points raised during the review process.

We look forward to receiving your revised manuscript.

Kind regards,

Kumar Venkitanarayanan, DVM, Ph.D.

Academic Editor

PLOS ONE

- https://www.tandfonline.com/doi/abs/10.1080/00071660902806962?journalCode=cbps20

- https://journals.plos.org/plosone/article?id=10.1371%2Fjournal.pone.0260923

In your revision ensure you cite all your sources (including your own works), and quote or rephrase any duplicated text outside the methods section. Further consideration is dependent on these concerns being addressed.

4. To comply with PLOS ONE submissions requirements, please provide methods of sacrifice in the Methods section of your manuscript.

Reviewers' comments:

Reviewer's Responses to Questions

**Comments to the Author**

1. Is the manuscript technically sound, and do the data support the conclusions?

Reviewer #1: Partly

Reviewer #2: No

2. Has the statistical analysis been performed appropriately and rigorously? 

Reviewer #1: No

Reviewer #2: No

3. Have the authors made all data underlying the findings in their manuscript fully available?

Reviewer #1: Yes

Reviewer #2: Yes

4. Is the manuscript presented in an intelligible fashion and written in standard English?

Reviewer #1: No

Reviewer #2: Yes

5. Review Comments to the Author

Reviewer #1: Summary: Interesting work. Please carefully review my comments and prepare a point-by-point response for the same. Revise the manuscript where appropriate. Thanks.

TITLE: Replace chicken with chickens.

Abstract:

Line 32: typo, please correct. Remove the extra “to”

Line 33: Please represent probiotic supplementation as percentage similar to chicory powder.

Line 37: IF you are representing antibiotic addition as gm/ton, please represent probiotic the same way. To avoid this confusion, I recommend representing your supplementations as percentage for all treatments.

Line 37: Please double check the abbreviation of gram. I believe it is “g”.

Introduction:

LINE 83: Remove the addition sign and replace with and.

Material and methods:

Line 108-113: Please include a table to describe the treatments clearly.

Line 116: Are the total bacterial load 32 billion CFU/100 g or each probiotic was at this concentration? Is this a commercial product or designed by the authors? How did the authors check the viability of each bacteria in the probiotic cocktail?

Line 120-122: Why are the authors presenting results in this section? I would recommend to move all the results presentation in the result section.

Line 118: Please represent probiotic levels as percentage too.

Line 130: “Feed intake”, please change to lower case F.

Line 134: Please include more details on how enzyme activity was determined. Describe how samples were stored, processed, followed by what type of assay was used.

Line 138: Please explain clearly how the 240 birds were sampled. It appears that only 1 bird/replicate that is 5 birds/treatment were processed for pH, intestinal contents etc. What about the other birds in the replicates? Why could the authors not process all birds for all samples to increase statistical power.

Question: How many trials/studies were conducted with 240 birds (6 dietary treatments, 8 replicates)?

Question: Where have the authors described the methodology for nutrient utilization? The results for nutrient utilization have been described in table 4 but I could not find the methodology portion for this section.

Question: How did the authors selectively enrich for Salmonella from the samples? What enrichment procedure was employed to facilitate recovery of Salmonella? Did the authors conduct a Salmonella load test at the start of the trials before the treatments began?

Results and Discussion:

Question: Why did the author not run groups for probiotic, chicory and coriander alone in the trials. Without these treatments, it is difficult to discern which component (probiotic, chicory, coriander) is effective in modulating body weight gains, FCR etc.

Table 7: Please share how the statistical analysis for comparing log CFU counts was conducted. I am interested to review log CFU/g values along with SE for each treatment. Some of the values for Lactobacillus (T5 7.88 vs T6 7.94), Salmonella (T4 7.33 vs T5 7.42) are very close to be statistically different from each other.

Reviewer #2: The authors investigated the effect of incorporating a combination of probiotics, chicory root powder and coriander seed powder on broiler growth performance, gut health and antioxidant activity. This study is relevant to the poultry industry for utilizing alternatives to antibiotic growth promoters.

However, my review for the manuscript was done partially because I felt that the experimental design and the number of birds used for the respective time points lacked clarity. It was hard to make an assessment on the data sets provided in the study. In addition, description provided for the statistical analysis and the inferences derived seemed incomplete. I would be happy to provide my further review and comments on the manuscript after the authors have provided a clarity on the experimental design and the N value for respective end points used in the study.

I have provided some of my comments for the manuscript below.

Abstract

Line 32-33 – please rephrase sentence as appropriate indicating the aim/background of the study. The details of the experimental design can be incorporated in the second sentence.

Introduction

Line76-77 – the connotation for antimicrobial properties of chicory root powder cannot be directly attributed but rather should be indicated as an indirect effect. Kindly rephrase the sentence appropriately. Or rather, if the authors had planned to indicate the antimicrobial activity other potential constituents, that may also be explained here.

Line 84 – the information regarding postbiotics may be removed from here. The authors can alternately provide additional references that cite the effect of probiotics + inulin combinations on poultry gut health. Specifically, the authors may cite the strains of probiotics that have been used by other researchers in a concise manner.

Line 90-92 – It would be better if the compounds listed in brackets are briefly expanded and described in terms of their chemical and functional properties/antimicrobial activities.

Materials and Methods

Line 116 – Does this mean the concentration of each bacterial strain or is this the total lactic acid bacterial counts in general?

Line 116 and Table 1 – Isn’t this analysis termed Proximate Analysis?

Line 118-121 – The way in which the treatment groups used for selecting the ideal formulation was not clear. What were the treatment groups involved in the broiler trials to determine the ideal composition of the mix that was eventually used for the final study?

Line 132 – clearly specify the N value for the metabolic trial.

Line 155-170 – how were the Salmonella colonies confirmed just by observed the colony morphology from the SS agar. What if the colonies observed in the plates were from Proteus?

Line 171 – was a blinded histopathological evaluation and analysis performed for the respective tissue samples?

6. PLOS authors have the option to publish the peer review history of their article (what does this mean?). If published, this will include your full peer review and any attached files.

Reviewer #1: No

Reviewer #2: No

---

## [Author Response · Author response to Decision Letter 0]

15 Mar 2022

Response to Reviewers

Editorial comments:

Reply: Followed the journal style

Reply: One of the publications was published by me, so there might be a chance of matching words as the methodology followed was the same. Duplicate reference in the reference section was identified and removed.

Reply: Corrected the manuscript for language usage, spelling, and grammar.

4. To comply with PLOS ONE submissions requirements, please provide methods of sacrifice in the Methods section of your manuscript.

Reply: Included in materials and methods

Reply: Raw data was included

Reply: Raw data was provided. Kindly update Data Availability statement.

Reviewer #1:

1. TITLE: Replace chicken with chickens.

 Reply by author: Modified as chickens

2. Line 32: typo, please correct. Remove the extra “to”

 Reply:Corrected

3. Line 33: Please represent probiotic supplementation as percentage similar to chicory powder.

Reply: Modified as per reviewer suggestion

4. Line 37: IF you are representing antibiotic addition as gm/ton, please represent probiotic the same way. To avoid this confusion, I recommend representing your supplementations as percentage for all treatments.

Reply:Expressed all additives in percentage

 5. Line 37: Please double check the abbreviation of gram. I believe it is “g”.

Reply: Modified as ‘g’

 Introduction:

LINE 83: Remove the addition sign and replace with and.

Reply: Removed typographical error

Material and methods:

 Line 108-113: Please include a table to describe the treatments clearly.

 -Included

Line 116: Are the total bacterial load 32 billion CFU/100 g or each probiotic was at this concentration? Is this a commercial product or designed by the authors? How did the authors check the viability of each bacteria in the probiotic cocktail?

Reply: The probiotic used is a multistrain probiotic trail product produced by Intron Biologicals, Hyderabad and it contains a total of 109 CFU/g of lyophilized and microencapsulated Bacillus coagulans, Saccharomyces boulardii, Lactobacillus acidophilus, Lactobacillus delbrueckii, Lactobacillus plantarum, Streptococcus thermophilus, Bacillus subtilis, Enterococcus faecium and Bifidobacterium bifidum. The viability was checked at the production itself.

Line 120-122: Why are the authors presenting results in this section? I would recommend to move all the results presentation in the result section.

Reply: Those lines were modified according to the materials and methods section. Removed lines 117 to 123.

Line 118: Please represent probiotic levels as percentage too.

Reply: Modified 

Line 130: “Feed intake”, please change to lower case F.

Reply: Corrected

Line 134: Please include more details on how enzyme activity was determined. Describe how samples were stored, processed, followed by what type of assay was used.

Reply: *Complete procedure of antoxidant enzyme activity has given below.

*ANTIOXIDANT RESPONSES 

 The antioxidant enzymes such as glutathione Peroxidase (GSHPx), glutathione Reductase (GSHRx) and superoxide Dismutase (SOD) were estimated by following the methods of Paglia and Valentine (1967), Carlberg and Mannervik (1985) and Madesh and Balsubramanian (1998) respectively. 

3.8.1 Glutathione Peroxidase (GSHPx) Enzyme Activity in Serum

 GSHPx activity was determined by the method proposed by Paglia and Valentine (1967) with slight modifications. Microtiter plates (96 well) were used to measure Glutathione peroxidase activity. To the 12.5 μl of serum, 250 μl of 0.1mM PBS (pH 7.4), 12.5 μl of H2O2 and 12.5 μl of reduced glutathione were added to wells and incubated at room temperature for 5 minutes, following which 12.5 μl of nicotinamide adenine dinucleotide phosphate (NADPH) solution was added and optical density was measured at 340 nm against the blank using ELISA reader - μQuant (BioTek instruments) for 5 minutes at 60 seconds interval and expressed as units/mg protein.

Glutathione Reductase (GSHRx) Enzyme Activity in Serum

 GSHRx activity was determined according to method described by Carlberg and Mannervik (1985) with slight modifications. Microtiter plates (96 well) were used to measure Glutathione reductase activity. To the 12.5 μl of serum, 250 μl of 0.1mM PBS (pH 7.4), 12.5 μl of oxidized glutathione, 12.5 μl of FAD and12.5 μl of 80 mM EDTA were added and incubated at room temperature for 15 minutes. Optical density was measured at 340 nm against the blank by using ELISA reader - μQuant (BioTek instruments) for 5 times at 60 seconds interval after addition of 12.5 μl of NADPH solution at last and expressed as units/mg protein. 

 Superoxide Dismutase (SOD) Enzyme Assay in Serum 

 Microtiter plates (96 well) were used for assay of SOD activity. To the 100 μl of test sample, 6 μl of 1.25 mM 3-(4,5-Dimethylthiazol-2-yl)-2,5-Diphenyltetrazolium Bromide (MTT) was added in duplicate for each sample. 15 μl of 100 μM pyrogallol and 29 μl of 25 mM PBS were added to make the volume to 150 μl. Pyrogollol was freshly prepared and added after the addition of all other reagents and incubated for 10 minutes at room temperature and the reaction was terminated with addition of 150 μl of dimethyl sulfoxide (DMSO), which arrests the reaction and dissolves the MTT formazan crystals formed. Plates were shaken well and optical density recorded at 570 nm using ELISA reader - μQuant (Madesh & Balsubramanian, 1998).

Line 138: Please explain clearly how the 240 birds were sampled. It appears that only 1 bird/replicate that is 5 birds/treatment were processed for pH, intestinal contents etc. What about the other birds in the replicates? Why could the authors not process all birds for all samples to increase statistical power.

Reply: One bird from each replicate means 8 birds for each treatment and total 48 samples were taken. According to the recommendations of IAEC, the number of birds to be sacrificed was limited as n=8 is enough for statistical analysis.

Question: How many trials/studies were conducted with 240 birds (6 dietary treatments, 8 replicates)? 

Ans: Only one trial was conducted with 240 birds.

Question: Where have the authors described the methodology for nutrient utilization? The results for nutrient utilization have been described in table 4 but I could not find the methodology portion for this section.

Ans: We did not mentioned complete procedure of metabolic trial in methodology, however, we followed the standard procedure recommended by AOAC (1997). The metabolic trial was conducted with one bird from each replicate to determine the retention efficiency of Dry Matter (DM), Crude Protein (CP) and energy as per the procedures described by AOAC (1997).

Question: How did the authors selectively enrich for Salmonella from the samples? What enrichment procedure was employed to facilitate recovery of Salmonella? Did the authors conduct a Salmonella load test at the start of the trials before the treatments began?

Ans: Eight birds from each dietary treatment were slaughtered on 42nd day and intestines were dissected at Meckel’s diverticulum. Approximately 5g of ileal digesta was collected aseptically into sterile sampling tubes and immediately transferred on ice to the laboratory for microbiological examination for E. coli, Salmonella spp and Lactobacilli spp counts. Salmonella-Shigella agar (SS Agar) for Salmonella spp. was used.

Then, 9 sterile test tubes with lids containing 9mL of phosphate buffer solution (PBS, pH-7.4) as diluent were prepared. Approximately 1g of the intestinal contents taken by sterile swab and homogenized for 3 minutes, aseptically mixed, added to the tubes, and diluted up to 109. Later, 1ml of the contents of each test tube was transferred to one of three selective agar media on petri plates, respectively (Gunal et al., 2006). Aerobic bacterial plates ( Salmonella spp.) were placed in an incubator at 37oC for 24 hours. Finally, the intestinal bacterial colony populations formed in each plate was counted by colony counter and the number of colonies was expressed as log10 value.

We did not conduct salmonella load test at the start of experiment as it was not required for our experiment.

Results and Discussion:

Question: Why did the author not run groups for probiotic, chicory and coriander alone in the trials. Without these treatments, it is difficult to discern which component (probiotic, chicory, coriander) is effective in modulating body weight gains, FCR etc.

Ans: Sir, we have conducted three separate experiments (other than this) with probiotic @ 10 g, 15 g, and 20 gm per 100kg, coriander @ 0.5%, 1.0% and 1.5% & chicory root powder @ 0.5%, 1.0% and 1.5%. The results of above experiments clearly indicated that each group from probiotic, chicory root powder and coriander treatments had shown desirable performance at particular dosages. Probiotic at 10g/100 kg, chicory at 1.0 % and coriander at 1.5 % levels significantly (P<0.05) increased the overall performance of broilers when compared to control, antibiotic and other levels of respective treatments.

 The above experimental results/data was already been published in Plosone and Indian journal of animal sciences. 

I am sharing the link of above article for your reference.

https://doi.org/10.1371/journal.pone.0260923

Table 7: Please share how the statistical analysis for comparing log CFU counts was conducted. I am interested to review log CFU/g values along with SE for each treatment. Some of the values for Lactobacillus (T5 7.88 vs T6 7.94), Salmonella (T4 7.33 vs T5 7.42) are very close to be statistically different from each other.

Ans: mentioned below

**E. coli counts (Calculated as per log10 colony forming units/gram of sample (106).

Treatment Replicates colonies cfu/ml (106 dilution0 log 10 value

T1 1 192 192000000 8.28

 1 201 201000000 8.30

 1 195 195000000 8.29

 1 189 189000000 8.28

 1 199 199000000 8.30

 1 188 188000000 8.27

 1 195 195000000 8.29

 1 200 200000000 8.30

T2 2 12 12000000 7.08

 2 14 14000000 7.15

 2 12 12000000 7.08

 2 15 15000000 7.18

 2 14 14000000 7.15

 2 15 15000000 7.18

 2 16 16000000 7.20

 2 16 16000000 7.20

T3 3 18 18000000 7.26

 3 16 16000000 7.20

 3 15 15000000 7.18

 3 15 15000000 7.18

 3 12 12000000 7.08

 3 14 14000000 7.15

 3 15 15000000 7.18

 3 16 16000000 7.20

T4 4 16 16000000 7.20

 4 19 19000000 7.28

 4 18 18000000 7.26

 4 25 25000000 7.40

 4 22 22000000 7.34

 4 23 23000000 7.36

 4 24 24000000 7.38

 4 25 25000000 7.40

T5 5 18 18000000 7.26

 5 32 32000000 7.51

 5 28 28000000 7.45

 5 29 29000000 7.46

 5 26 26000000 7.41

 5 28 28000000 7.45

 5 27 27000000 7.43

 5 25 25000000 7.40

T6 6 16 16000000 7.20

 6 14 14000000 7.15

 6 15 15000000 7.18

 6 16 16000000 7.20

 6 18 18000000 7.26

 6 13 13000000 7.11

 6 17 17000000 7.23

 6 16 16000000 7.20

Reviewer #2:

- Abstract

Line 32-33 – please rephrase sentence as appropriate indicating the aim/background of the study. The details of the experimental design can be incorporated in the second sentence.

Reply: Included as per recommendations.

Introduction

Line76-77 – the connotation for antimicrobial properties of chicory root powder cannot be directly attributed but rather should be indicated as an indirect effect. Kindly rephrase the sentence appropriately. Or rather, if the authors had planned to indicate the antimicrobial activity other potential constituents, that may also be explained here.

Reply: Modified according to the recommendations

Line 84 – the information regarding postbiotics may be removed from here. The authors can alternately provide additional references that cite the effect of probiotics + inulin combinations on poultry gut health. Specifically, the authors may cite the strains of probiotics that have been used by other researchers in a concise manner.

Reply: Incorporated accordingly.

Line 90-92 – It would be better if the compounds listed in brackets are briefly expanded and described in terms of their chemical and functional properties/antimicrobial activities.

Reply: Sir, They were already proven compounds as indicated by reference, no need to discuss in detail.

Line 116 – Does this mean the concentration of each bacterial strain or is this the total lactic acid bacterial counts in general?

Reply: Total bacterial count (Bacillus coagulans, saccharomyces boulardii, Lactobacillus acidophilus, Lactobacillus delbrueckii, Lactobacillus plantarum, Streptococcus thermophilus, Bacillus subtilis, Enterococcus faecium, Bifidobacterium bifidum).

Line 116 and Table 1 – Isn’t this analysis termed Proximate Analysis?

Reply: Yes sir, proximate analysis.

Line 118-121 – The way in which the treatment groups used for selecting the ideal formulation was not clear. What were the treatment groups involved in the broiler trials to determine the ideal composition of the mix that was eventually used for the final study?

Reply: Sir, we have conducted three separate experiments (other than this) with probiotic @ 10 g, 15 g, and 20 gm per 100kg, coriander @ 0.5%, 1.0% and 1.5% & chicory root powder @ 0.5%, 1.0% and 1.5%. The results of above experiments clearly indicated that each group from probiotic, chicory root powder and coriander treatments had shown desirable performance at particular dosages. Probiotic at 10g/100 kg, chicory at 1.0 % and coriander at 1.5 % levels significantly (P<0.05) increased the overall performance of broilers when compared to control, antibiotic and other levels of respective treatments.

 The above experimental results/data was already been published in Plosone and Indian journal of animal sciences. 

I am sharing the links of above articles for your reference.

https://doi.org/10.1371/journal.pone.0260923

Line 132 – clearly specify the N value for the metabolic trial.

Reply: N value was already mentioned in table no 4. 

The metabolic trial was conducted with one bird from each replicate means 8 birds treatment and total 48 samples were used for estimation of Dry Matter (DM), Crude Protein (CP) and energy.

Line 155-170 – how were the Salmonella colonies confirmed just by observed the colony morphology from the SS agar. What if the colonies observed in the plates were from Proteus?

Reply: Translucent colonies with black centers which are typical to Salmonella on SS agar were counted by using colony counter and the number of colonies was expressed as log10 value.

 The complete procedure was explained below.

Eight birds from each dietary treatment were slaughtered on 42nd day and intestines were dissected at Meckel’s diverticulum. Approximately 5g of ileal digesta was collected aseptically into sterile sampling tubes and immediately transferred on ice to the laboratory for microbiological examination for E. coli, Salmonella spp and Lactobacilli spp counts. Eosin methylene blue agar (EMB) for E. coli growth, Salmonella-Shigella agar (SS Agar) for Salmonella spp. and MRS agar (De Man, Rogosa and Sharpe agar) for Lactobacilli spp growth was used.

Then, 9 sterile test tubes with lids containing 9mL of phosphate buffer solution (PBS, pH-7.4) as diluent were prepared. Approximately 1g of the intestinal contents taken by sterile swab and homogenized for 3 minutes, aseptically mixed, added to the tubes, and diluted up to 109. Later, 1ml of the contents of each test tube was transferred to one of three selective agar media on petri plates, respectively (Gunal et al., 2006). Aerobic bacterial plates (E. coli, Salmonella spp.) were placed in an incubator at 37oC for 24 hours. Anaerobic (Lactobacilli spp.) medium plates were placed in an anaerobic jar with an anaerobic gas pack system at 37oC for 24 hours. Finally, the intestinal bacterial colony populations formed in each plate was counted by colony counter and the number of colonies was expressed as log10 value.

Line 171 – was a blinded histopathological evaluation and analysis performed for the respective tissue samples?

Reply: Yes sir, the HP slides were analyzed blindly, except the control group, so as to have a reference point. We have taken 8 histopathological sections per treatment each from each bird.

---

## [Editor Report · Decision Letter 1]

28 Apr 2022

PONE-D-21-40040R1Synergistic effect of probiotic, chicory root powder and coriander seed powder on growth performance, antioxidant activity and gut health of broiler chickenPLOS ONE

Dear Dr. Gurram,

Thank you for submitting your manuscript to PLOS ONE. After careful consideration, we feel that it has merit but does not fully meet PLOS ONE’s publication criteria as it currently stands. Therefore, we invite you to submit a revised version of the manuscript that addresses the points raised during the review process.

We look forward to receiving your revised manuscript.

Kind regards,

Kumar Venkitanarayanan, DVM, Ph.D.

Academic Editor

PLOS ONE

Journal Requirements:

Additional Editor Comments (if provided):

The following changes need to be done.

Lines 48-51 : Revise the sentence. Significantly appears twice in the sentence.

Lines 75-77: Recently, herbal feed additives products like chicory root powder are gaining attention as they indirectly promote antimicrobial 77 action by reducing the harmful bacteria in the gut. Replace “feed additives” with “feed additive”.

Confirm on the manuscript that all the treatment concentrations in the feed were weight/weight basis. If not, please indicate.

Lines 125-127: The dose levels (Probiotic at 10g/100 kg, chicory at 1.0 % and coriander at 1.5 % levels ) were selected based on my previous trail results. Delete “my”.

Line 368: Significant increase in the counts of Lactobacilli in test diets was also support the above results. Delete “was”.

Lines 503-511: Suggest revising the following text as below.

Supplementation of probiotic + chicory, probiotic + coriander, chicory + coriander and probiotic + chicory + coriander combinations produced greater weight gain, improved FCR, and higher antioxidant activity compared to control and antibiotic at 42 d of age. The combination of probiotic (0.01%) with chicory root powder (1.0%) was more effective than combinations of other additives in terms of body weight gain and FCR. Supplementation of different combinations of probiotic, chicory root powder and coriander seed powder significantly lowered the gut pH, E. coli and Salmonella counts and increased the Lactobacilli counts. In addition, this treatment improved the gut morphometry parameters such as VH, CD and VW in the in the intestines. Thus, supplementation of probiotic at 0.01%, chicory root powder at 1.0 %, and coriander seed powder at 1.5 % combinations could be used in the diet as a potential growth promoter in broiler chickens. However, follow up large-scale studies under field conditions are necessary before recommending the compounds in the broiler diet.

---

## [Author Response · Author response to Decision Letter 1]

2 May 2022

Synergistic effect of probiotic, chicory root powder and coriander seed powder on growth performance, antioxidant activity and gut health of broiler chicken

1. Please ensure that you refer to Table 1 in your text as, if accepted, production will need this reference to link the reader to the Table.

Response: Table 1 has been included in text and please provide reference link to Table 1.

2. Please amend the title either on the online submission form or in your manuscript so that they are identical.

Response: corrected

3. Please provide additional details regarding participant consent. In the Methods section, please ensure that you have specified (1) whether consent was informed and (2) what type you obtained (for instance, written or verbal). If your study included minors, state whether you obtained consent from parents or guardians. If the need for consent was waived by the ethics committee, please include this information.

Response: No consent was raised by animal ethics committee while obtaining permission. Moreover, the consent was not required for conducting experiments in Broilers. The same has been included in ethics statement.

---

## [Editor Report · Decision Letter 2]

7 Jun 2022

Synergistic effect of probiotic, chicory root powder and coriander seed powder on growth performance, antioxidant activity and gut health of broiler chickens

PONE-D-21-40040R2

Dear Dr. Gurram,

We’re pleased to inform you that your manuscript has been judged scientifically suitable for publication and will be formally accepted for publication once it meets all outstanding technical requirements.

Kind regards,

Kumar Venkitanarayanan, DVM, Ph.D.

Academic Editor

PLOS ONE

---

## [Editor Report · Acceptance letter]

16 Jun 2022

PONE-D-21-40040R2 

*Synergistic effect of probiotic, chicory root powder and coriander seed powder on growth performance, antioxidant activity and gut health of broiler chickens*

Dear Dr. Gurram:

I'm pleased to inform you that your manuscript has been deemed suitable for publication in PLOS ONE. Congratulations! Your manuscript is now with our production department. 

Kind regards, 

on behalf of

Dr. Kumar Venkitanarayanan 

Academic Editor

PLOS ONE